# Avoidance of stochastic RNA interactions can be harnessed to control protein expression levels in bacteria and archaea

Sinan Uğur Umu[1,2], Anthony M Poole[1,2], Renwick CJ Dobson[1,2,3], Paul P Gardner[1,2,4]*

[1]School of Biological Sciences, University of Canterbury, Christchurch, New Zealand; [2]Biomolecular Interaction Centre, University of Canterbury, Christchurch, New Zealand; [3]Department of Biochemistry and Molecular Biology, University of Melbourne, Parkville, Australia; [4]BioProtection Research Centre, University of Canterbury, Christchurch, New Zealand

**Abstract** A critical assumption of gene expression analysis is that mRNA abundances broadly correlate with protein abundance, but these two are often imperfectly correlated. Some of the discrepancy can be accounted for by two important mRNA features: codon usage and mRNA secondary structure. We present a new global factor, called mRNA:ncRNA avoidance, and provide evidence that avoidance increases translational efficiency. We also demonstrate a strong selection for the avoidance of stochastic mRNA:ncRNA interactions across prokaryotes, and that these have a greater impact on protein abundance than mRNA structure or codon usage. By generating synonymously variant green fluorescent protein (GFP) mRNAs with different potential for mRNA:ncRNA interactions, we demonstrate that GFP levels correlate well with interaction avoidance. Therefore, taking stochastic mRNA:ncRNA interactions into account enables precise modulation of protein abundance.

*For correspondence: paul.gardner@canterbury.ac.nz

**Competing interests:** The authors declare that no competing interests exist.

## Introduction

It should in principle be possible to predict protein abundance from genomic data. However, protein and mRNA levels are not strongly correlated (*de Sousa Abreu et al., 2009*; *Vogel and Marcotte, 2012*; *Kwon et al., 2014*; *Maier et al., 2011*; *Lu et al., 2007*; *Taniguchi et al., 2010*; *Chen et al., 2016*), which is a major barrier to precision bioengineering and quantification of protein levels. mRNA secondary structure (*Pelletier and Sonenberg, 1987*; *Chamary and Hurst, 2005*), codon usage (*Ikemura, 1981*; *Sharp and Li, 1987*; *Andersson and Kurland, 1990*), and mRNA (and protein) degradation rates (*Maier et al., 2011*) are commonly invoked to explain this discrepancy (*Boël et al., 2016-21*). Yet, at best, these features account for only 40% of variation, and in some instances explain very little of the observed variation (*Kudla et al., 2009*; *Maier et al., 2011*; *Plotkin and Kudla, 2011*; *Goodman et al., 2013*; *Chen et al., 2016*). Here we show that crosstalk interactions between ncRNAs and mRNAs also impact protein abundance, and that such interactions have a greater effect than either mRNA secondary structure or codon usage. We measured interactions between a set of evolutionarily conserved core mRNAs and ncRNAs from 1700 prokaryotic genomes using minimum free energy (MFE) models. For 97% of species, we find a reduced capacity for interaction between native RNAs relative to controls. Furthermore, by generating synonymously variant GFP mRNAs that differ in their potential to interact with core ncRNAs, we demonstrate that GFP expression levels can be both predicted and controlled. Our results demonstrate that there is strong selection for the avoidance of stochastic mRNA:ncRNA interactions across prokaryotes.

**eLife digest** Many genes carry information for making proteins. To make a protein, a working copy of the information stored in DNA is first copied into a molecule of messenger RNA. These RNA messages are then interpreted by the ribosome, the molecular machine that makes proteins. Many messages are produced from each gene, and each message can be read multiple times. Thus, it should follow that the number of messages produced dictates the number of proteins made. However, this is not the case and the number of proteins produced cannot be completely predicted from knowing the number of messenger RNAs.

Cells control how much of a given protein they produce through interactions between the messenger RNAs and other regulatory RNAs. The regulatory RNAs bind directly to a message and impede protein production. Because there are millions of RNAs in a cell, these interactions have evolved to be highly specific. Nevertheless, it seems inevitable that messenger RNAs would encounter other RNAs too, which could short-circuit gene regulation and lead to less protein being produced.

Umu et al. have now asked if such short-circuit events are selected against during evolution. Computational tools were used to predict the strength of binding between the RNAs found in the dominant forms of microbial life on Earth: the bacteria and the archaea. This approach revealed that the majority of messenger RNAs bind more weakly to the most common RNA molecules found in cells than would be expected by chance. Weakened binding should prevent the RNA molecules from becoming tangled with each other and ensure that protein levels are not perturbed by unintended interactions between highly expressed messages and other RNAs.

To test this hypothesis further, Umu et al. generated versions of the gene for a green fluorescent protein that differed only in how well their messenger RNAs could avoid interacting with the most abundant RNAs in *E. coli* cells. Those messengers that were designed to avoid interacting with other RNAs yielded far more protein than those that were not. The findings show that taking this kind of avoidance into account can improve predictions about how much protein will be produced and should therefore make it easier to control protein production in experimental systems.

Finally, the messenger RNAs of some bacteria do not show such clear avoidance. However, these bacteria have a more complex internal cell structure. This finding hints at an alternative means for avoiding short-circuiting events that could be used by more complicated cells, such of those of animals and plants, which also contain much larger numbers of RNAs.

Applying this knowledge to mRNA design will enable precise control of protein abundance through the incorporation or exclusion of inhibitory interactions with native ncRNAs.

## Results and discussion

To examine if avoidance of stochastic mRNA:ncRNA interactions is a feature of transcriptomes in bacteria and archaea, we estimated the strength of all possible intermolecular RNA interactions using a minimum free energy (MFE) model (*Mückstein et al., 2006*) using core ncRNAs and mRNAs. In this work the core ncRNAs are six well conserved and highly expressed tRNA, rRNA, RNase P RNA, SRP RNA, tmRNA and 6S RNA families annotated by Rfam (*Gardner et al., 2011*; *Nawrocki et al., 2015*), the core mRNAs are 114 well conserved mRNAs found across bacteria, 40 of which are also conserved across archaea (*Wu et al., 2013*).

If stochastic interactions are selected against, because of the capacity for abundant ncRNAs (*Lindgreen et al., 2014*; *Deutscher, 2006*; *Giannoukos et al., 2012*) to impact translation (*Waters and Storz, 2009*; *Storz et al., 2011*), such negative selection would be most comparable between species and readily detected for broadly conserved ncRNAs and mRNAs. Under-representation of interactions has been considered for the specific case of Shine-Dalgarno-like (SD-like) sequences and the ribosome (*Li et al., 2012*; *Woolstenhulme et al., 2015*; *Borg and Ehrenberg, 2015*; *Diwan and Agashe, 2016*) and between microRNAs and 3' UTRs (*Bartel and Chen, 2004*; *Farh, 2005*; *Stark et al., 2005*; *van Dongen et al., 2008*). We computed the free energy distribution of interactions between highly conserved mRNA:ncRNA pairs and compared this to a number

of negative control interactions, which serve to show the expected distribution of binding energy values (*Figure 1A*). The initiation of translation has been shown to be the rate limiting step for translation (*Tuller and Zur, 2015*; *Plotkin and Kudla, 2011*; *Nakahigashi et al., 2014*), therefore, we focus our analysis on the first 21 nucleotides of the mRNA coding sequence (CDS). This has the further advantage of reducing computational complexity. We also test a variety of negative control mRNA regions, which are unlikely to play a functional role in RNA:ncRNA interactions. The mRNA controls include (1) di-nucleotide preserving shuffled sequences (*Workman, 1999*) (orange, *Figure 1A*), (2) homologous mRNAs from another phylum (with a compatible guanine-cytosine (G +C) content) (purple), (3) downstream regions 100 base pairs (bps) within the CDS (pink), (4) the reverse complement of the 5′ of CDSs (green), and lastly (5) unannotated (intergenic) genomic regions (yellow). Our interaction predictions in a single model strain show that native interactions consistently have higher (i.e. less stable) free energies than expected when compared to the five different mRNA negative controls: that is, there is a reduced capacity for native mRNAs and native ncRNAs to interact. We also compared different energy models and confirm that the MFE shift is a result of intermolecular binding (*Figure 1—figure supplement 1A–C*). We subsequently deployed the most conservative negative control (i.e. di-nucleotide preserving shuffle) and free energy model (*Figure 1—figure supplement 1C*) to detect if this shift for less stable binding of mRNA:ncRNA is true of all bacteria and archaea.

In terms of stoichiometry, the model we use assumes that ncRNA expression levels are vastly in excess of mRNA expression levels (i.e. [ncRNA] >> [mRNA]) (*Giannoukos et al., 2012*; *Deutscher, 2006*). This is generally a biologically reasonable assumption when focussing on core genes based upon past analysis and our own work with RNA-seq data from a range of bacteria and archaea (Figure 4) (*Lindgreen et al., 2014*). Consequently, any potential mRNA interaction regions are saturated with ncRNA, therefore a summative model of interaction energies is a reasonable approximation to the estimated impact of excess hybridization. If modelling ncRNAs that are not so abundant, then a model weighted by expression level may be advantageous, but it is difficult to assess these across all conditions and developmental stages that are evolutionarily relevant. In order to ensure that our analysis is comparable across all bacteria and archaea we have focussed on just the most highly conserved ncRNA and protein-coding genes. Although, many of the ncRNAs are highly structured and are bound by RNA-binding proteins this is not the case during either synthesis and degradation of these products, furthermore, a fraction of the RNA components of these genes will be exposed. Therefore, we expect these will form useful datasets for initial testing of our hypothesis.

In order to assess whether mRNA:ncRNA avoidance is an evolutionarily conserved phenomenon, we calculated intermolecular binding energies for conserved ncRNAs and mRNAs from 1,582 bacterial and 118 archaeal genomes and compared these to a negative control dataset derived using a di-nucleotide frequency preserving shuffling procedure (*Workman, 1999*). This measures a property that we call the 'extrinsic avoidance' of mRNA:ncRNA interactions, yet this approach may fail to identify genuine avoidance in cases when the G+C content differences between interacting RNAs is extreme. Measuring only extrinsic avoidance (using shuffled mRNAs as negative controls), we found that stochastic mRNA:ncRNA interactions are significantly underrepresented in most (73%) of the prokaryotic phyla (p<0.05, one-tailed Mann-Whitney U test) (*Figure 1B,C* and *Figure 1—figure supplement 2*). This indicates that there is selection against stochastic interactions in both bacteria and archaea.

We next sought to establish the degree to which intrinsic G+C features of RNAs lead to avoidance of stochastic interactions (*Figure 1D*). A similar idea has been proposed which suggests that purine loading in thermophilic bacteria may limit mRNA:mRNA interactions (*Lao and Forsdyke, 2000*). A test of G+C composition revealed a significant difference (p<0.05, two-tailed Mann-Whitney U test) between mRNAs and ncRNAs for 95% of bacteria and archaea (*Figure 1D,E*). Therefore, either extrinsic or intrinsic avoidance signals indicate that selection against stochastic interactions and it is near-universal for the prokaryotes (97% of all strains) (*Figure 1E* and *Supplementary file 1A and B*).

Our results clearly establish a signature of selection that acts to minimise stochastic mRNA:ncRNA interactions. However, with thousands of potential interacting RNA species in even simple prokaryotic systems (*Vivancos et al., 2010*; *Sharma et al., 2010*), the complete avoidance of stochastic interactions is combinatorially unlikely. Therefore, there ought to be a tradeoff between avoidance

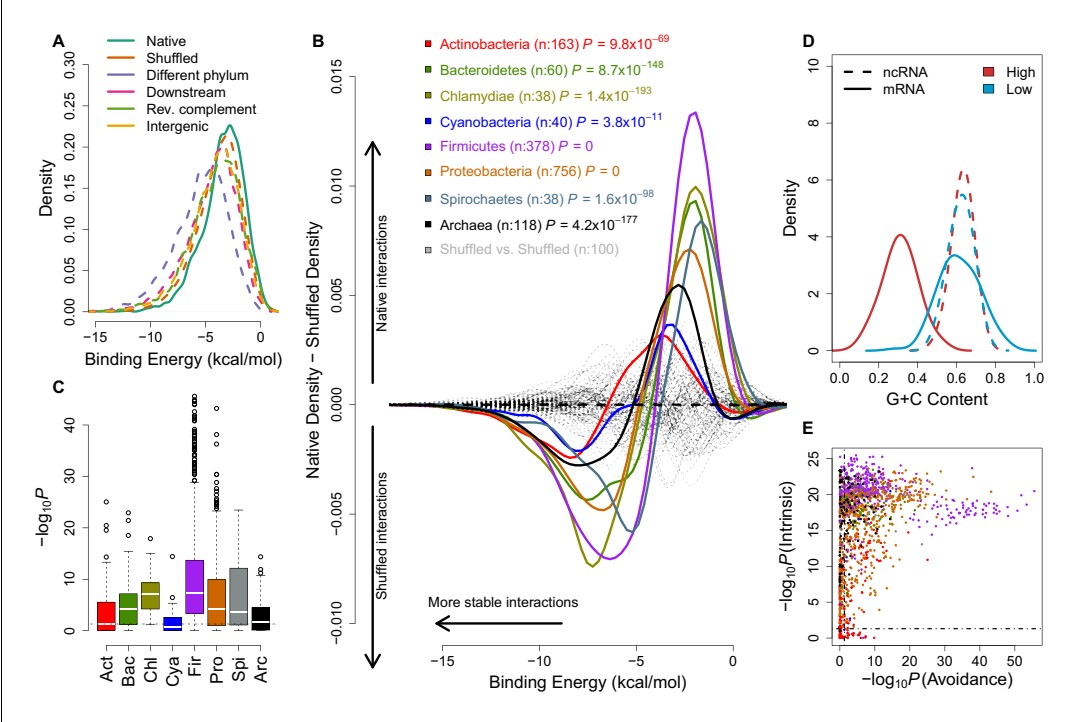

**Figure 1.** mRNA:ncRNA avoidance is a conserved feature of bacteria and archaea. (**A**) Native core mRNA:ncRNA binding energies (green line; mean = −3.21 kcal/mol) are significantly higher than all mRNA negative control binding energies (dashed lines; mean binding energies are -3.62, -5.21, -4.13, -3.86 & -3.92 kcal/mol respectively) in pairwise comparisons (p<2.2 × $10^{-16}$ for all pairs, one-tailed Mann-Whitney U test) for *Streptococcus suis* RNAs. (**B**) The difference between the density distributions of native mRNA:ncRNA binding energies and dinucleotide preserved shuffled mRNA:ncRNA controls as a function of binding energy for different taxonomic phyla. Each coloured curve illustrates the degree of extrinsic avoidance for different bacterial phyla or the archaea. Positive differences indicate an excess in native binding for that energy value, negative differences indicate an excess of interactions in the shuffled controls. The dashed black line shows the expected result if no difference exists between these distributions and the dashed grey lines show empirical differences for shuffled vs shuffled densities from 100 randomly selected bacterial strains. (**C**) This box and whisker plot shows −$\log_{10}(P)$ distributions for each phylum and the archaea, the p-values are derived from a one-tailed Mann-Whitney U test for each genome of native mRNA:ncRNA versus shuffled mRNA:ncRNA binding energies. The black dashed line indicates the significance threshold (p<0.05). (**D**) A high intrinsic avoidance strain (*Thermodesulfobacterium sp*. OPB45) shows a clear separation between the G+C distribution of mRNAs and ncRNAs (p=9.2 × $10^{-25}$, two-tailed Mann-Whitney U test), and a low intrinsic avoidance strain (*Mycobacterium sp*. JDM601) has no G+C difference between mRNAs and ncRNAs (p=0.54, two-tailed Mann-Whitney U test). (**E**) The x-axis shows −$\log_{10}(P)$ for our test of extrinsic avoidance using binding energy estimates for both native and shuffled controls, while the y-axis shows −$\log_{10}$(P) for our intrinsic test of avoidance based upon the difference in G+C contents of ncRNAs and mRNAs. Two perpendicular dashed black lines show the threshold of significance for both avoidance metrics. 97% of bacteria and archaea are significant for at least one of these tests of avoidance.

The following figure supplements are available for figure 1:

**Figure supplement 1.** Applying different energy models of intramolecular and intermolecular interactions for native sequences and various negative controls.

**Figure supplement 2.** The top and the bottom panels show bacterial phyla and archaeal phyla respectively.

and optimal expression. To assess this, we examined the relationship between potential stochastic interactions and the variation between mRNA and cognate protein levels for four previously published endogenous mass spectrometry datasets from *Escherichia coli* (*E. coli*) and *Pseudomonas aeruginosa* (*P. aeruginosa*) (**Laurent et al., 2010**; **Kwon et al., 2014**; **Lu et al., 2007**). We computed Spearman's correlation coefficients between protein abundances and extrinsic avoidance, 5′ end internal mRNA secondary structure and codon usage. Of the three measures, avoidance is significantly correlated in all four datasets (Spearman's rho values are between 0.11–0.17 and corresponding p-values are between 0.01 and 1.3 × $10^{-12}$). In contrast, 5′ end mRNA structure significantly correlates in two datasets, and codon usage significantly correlates in all four datasets. This indicates

that, despite strong selection against stochastic interactions, such interactions do significantly impact the proteome (*Figure 2A* and *Supplementary file 3*). We have also conducted an 'outlier analysis' on one of the *E. coli* datasets (*Laurent et al., 2010*). We have selected the top and bottom-most expressed genes relative to mRNA expression levels and computed Z-scores for each of codon-usage, internal secondary structure and avoidance measures. We found that avoidance measures show the most extreme shifts downwards for the bottom-most expressed genes and is shifted the highest for the top-most genes (*Figure 2—figure supplement 4*).

We also test how mRNA:ncRNA crosstalk impacts the translation of transformed mRNAs that have not coevolved with the ncRNA repertoire (low avoidance mRNAs are rare in native datasets). We examined two available *E. coli*-based GFP experimental datasets (*Goodman et al., 2013*; *Kudla et al., 2009*), where synonymous mRNAs are generated for a GFP reporter gene. This enables the assessment of the impact of synonymous changes on protein abundance using fluorescence. Avoidance and mRNA secondary structure are both significantly correlated with fluorescence, whereas codon usage is not (Spearman's rho values are 0.11 and 0.65, the corresponding p-values are $3.17 \times 10^{-41}$ and $1.69^{-20}$) (*Figure 2A*). Note that one of the GFP datasets (*Goodman et al., 2013*) uses native *E. coli* mRNA 5′ ends for their constructs, whereas the other GFP dataset (*Kudla et al., 2009*) is randomly generated. We observe that the influence of avoidance on gene expression for randomly sampled synonymous mRNAs is strong (*Figure 2—figure supplement 3*), while endogenous gene expression is limited. Presumably, due to negative selection pruning low avoidance mRNAs from the gene pool (*Figure 2A*).

For each of the seven datasets described above we have tested linear models of measures of mRNA levels, codon usage, internal secondary structure and avoidance (*Figure 2—figure supplement 3* and *Supplementary file 5*). Avoidance alone explains around 35% of variance in GFP datasets where extreme mRNA compositions can be explored, whereas in native mass-spec derived datasets 2–3% of the variance is explained by avoidance alone. Codon usage describes 2% to −0.5% of variance in GFP data, and 19% to 0.3% of variance in mass-spec derived datasets. Internal secondary structure 33% to 10% in GFP datasets, and 0.2% to 0% of the variance in mass-spec derived datasets. Using all four measures in combination across the seven datasets between 70% and 42% of variation in protein levels can be explained, removing avoidance from the model reduces these estimates by between 56% and 0.7%. Thus, avoidance is at least as good an explanation of variation in protein abundance as either codon usage and internal mRNA secondary structure.

Our results indicate that crosstalk between mRNAs and ncRNAs can impact protein expression levels. We therefore predict that taking crosstalk into account will enable the design of constructs where protein expression levels can be precisely controlled. To test this, we generated GFP constructs based on the following constraints: codon bias, 5′ end mRNA secondary structure stability and crosstalk avoidance (see Materials and methods). Our constructs are designed to capture the extremes of one variable, while controlling other variables (e.g. high or low avoidance and near-average codon bias and mRNA secondary structure). The G+C content, a known confounding factor, was also strictly controlled for each construct. We selected a commercial service to perform our GFP transformations to avoid possible bias and increase the robustness of our approach (*Ioannidis and Khoury, 2011*). We predicted that a construct where all three parameters are optimised will result in a higher expression. Consistent with predictions, our optimised construct had maximal expression (*Figure 2—figure supplement 1*). Of the three parameters, avoidance showed the largest range, suggesting that tuning this parameter permits expression levels to be finely controlled ($R_s$ = 0.56, p=$6.9 \times 10^{-6}$) (*Figure 2B–D* and *Figure 2—figure supplements 1–4*).

For a final confirmation of the avoidance hypothesis, we tested the *Thermus thermophilus* (*T. thermophilus*) HB8 SSU ribosomal RNA, which is a component of one of the most complete prokaryotic ribosomal structures available in the PDB (*Rozov et al., 2015*). We identified the regions of the SSU rRNA that had the least capacity to interact with *T. thermophilus* core mRNAs and found that these regions were generally not bound to either ribosomal proteins or other ncRNAs, such as the LSU rRNA (p=$2.49 \times 10^{-17}$, Fisher's exact test) (*Figure 3*; see 'Materials and methods'). The influence of internal SD-like regions on translation pausing have been described elsewhere (*Li et al., 2012*), in addition we note that the anti-SD region on SSU rRNA is one of the RNA avoidance regions (*Figure 3A*).

This study focusses on the 5 ′ ends of the CDS as this region is important for the initiation of translation (*Plotkin and Kudla, 2011*; *Tuller and Zur, 2015*) and is a consistent feature of all the

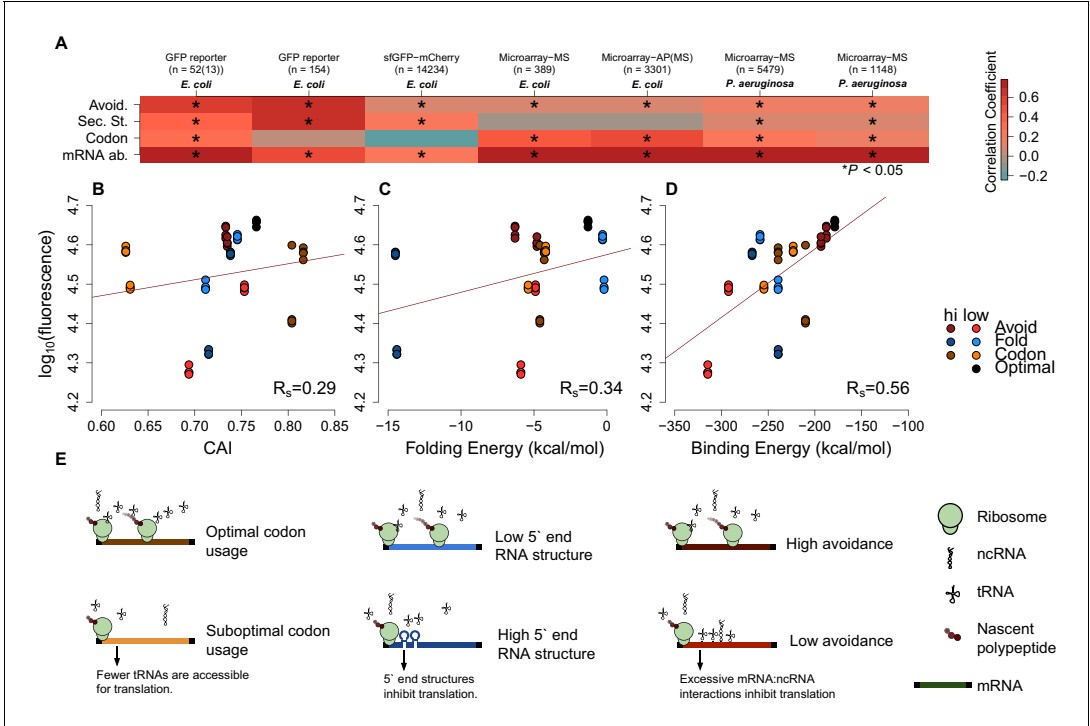

**Figure 2.** mRNA attributes have different impacts on protein abundance. (**A**) This heatmap summarizes the effect sizes of four mRNA attributes (avoidance of mRNA:ncRNA interaction, 5′ end secondary structure, codon bias and mRNA abundance) on protein expression as Spearman's correlation coefficients, which are represented in gradient colors, while a starred block shows if the associated correlation is significant (p<0.05). (**B**) GFP expression correlates with optimized codon selection, measured by CAI ($R_s$ = 0.29, p=0.016). (**C**) GFP expression correlates with 5′ end secondary structure of mRNAs, measured by 5′ end intramolecular folding energy ($R_s$ = 0.34, p=0.006). (**D**) GFP expression correlates with avoidance, measured by mRNA:ncRNA binding energy ($R_s$ = 0.56, p=6.9 × $10^{-6}$). (**E**) Each cartoon illustrates the corresponding hypothesis; (1) optimal codon distribution (corresponding tRNAs are available for translation), (2) low 5′ end RNA structure (high folding energy of 5′ end) and (3) avoidance (fewer crosstalk interactions) lead to faster translation.

The following figure supplements are available for figure 2:

**Figure supplement 1.** GFP mRNA constructs have an unbiased design that produces different protein expressions.

**Figure supplement 2.** The scatter-plots of protein abundances (as log-fluorescences) summarize the effect of general factors for extreme GFP and previously published GFP datasets.

**Figure supplement 3.** In the lower four panels we show the $R^2$ values for linear regression models between measures of each of avoidance, internal secondary structure, codon usage and mRNA levels for each of seven independent protein and mRNA expression datasets *Supplementary file 5*).

**Figure supplement 4.** An outlier analysis of E. coli protein-per-mRNA ratios and avoidance, codon usage and internal mRNA secondary structure statistics.

**Figure supplement 5.** Overview of mRNA:ncRNA avoidance analysis and results.

genomic, transcriptomic, proteomic and GFP expression datasets that we have evaluated in this work. In smaller-scale tests we have observed similar conserved avoidance signals within the entire CDSs (*Figure 3—figure supplement 1*) and within the 5′UTRs (*Figure 3—figure supplement 2*). Furthermore, we predict that similar signals can be observed for mRNA:mRNA and ncRNA:ncRNA avoidance. Although the impacts of these features are challenging to validate, interactions between clustered regularly interspaced short palindromic repeats (CRISPR) spacer sequences (*Bhaya et al., 2011*) and core ncRNAs are good candidates to test ncRNA:ncRNA avoidance.

In conclusion, our results indicate that the specificity of prokaryotic ncRNAs for target mRNAs is the result of selection both for a functional interaction and against stochastic interactions. Our experimental results support the view that stochastic interactions are selected against, due to deleterious outcomes on expression. We suspect avoidance of crosstalk interactions has several evolutionary consequences. First, as transcriptional outputs become more diverse in evolution, we expect that the probability of stochastic interactions for both new ncRNAs and mRNAs becomes higher. This will impact the emergence of new, high abundance RNAs, since selection for high abundance may be mitigated by deleterious crosstalk events. Second, we predict that stochastic interactions limit the number of simultaneously transcribed RNAs, since the combinatorics of RNA:RNA interactions imply that eventually stochastic interactions cannot be avoided. This may in turn drive selection for forms of spatial or temporal segregation of transcripts. Finally, taking codon usage, mRNA secondary structure and potential mRNA:ncRNA interactions into account allows better prediction of proteome outputs from genomic data, and informs the precise control of protein levels via manipulation of synonymous mRNA sequences (*Figure 2—figure supplement 5*).

## Materials and methods

Here we summarize the data sources, materials and methods corresponding to our manuscript. We performed all statistical analyses in R, and all other computational methods in Python 2.7 or Bash shell scripts. We explicitly cite all the bioinformatics tools and their versions. All tables (*Supplementary files 1–5*) are available as supporting online material. All of our own sequences, scripts and R workspace images are available on Github including the supplementary files (http://github.com/UCanCompBio/Avoidance). The other datasets are cited in the manuscript (*Supplementary file 3*).

### Evolutionary conservation

If excessive interactions between messenger RNAs (mRNAs) and non-coding RNAs (ncRNAs) are detrimental to cellular function, then we expect the signature of selection against interactions (avoidance) to be a conserved feature of prokaryotic genomes. In the following, we describe where the data used to test the evolutionary conservation of avoidance was acquisitioned, the models that we use to test avoidance and the negative controls in detail for evolutionary conservation predictions. We also investigate detect regions of avoidance on one of the core ncRNAs, the ribosomal small subunit (SSU) RNA.

### Data sources for bacterial genomes

The bacterial genomes and annotations that we used for investigating mRNA:ncRNA interactions were acquired from the EBI nucleotide archive (2564 sequenced bacterial genomes available on August 2013; http://www.ebi.ac.uk/genomes/bacteria.html). We selected an evolutionarily conserved (core) group of 114 mRNAs from PhyEco (*Wu et al., 2013*) and an evolutionarily conserved (core) group of ncRNAs (*Hoeppner et al., 2012*). PhyEco markers are based on a set of profile HMMs that correspond to highly conserved bacterial protein coding genes (these include ribosomal proteins, tRNA synthetases as well as other components of translation machinery, DNA repair and polymerases) (*Wu et al., 2013*). The HMMer package (version 3.1b1) (*Eddy, 2011*) was used to extract the mRNAs corresponding to these marker genes from genome files. We removed genome sequences that host fewer than 90% of the marker genes; leaving 1582 bacterial genome sequences and 176,704 core mRNAs that spanned these. We extracted the first to the 21st nucleotide of the core mRNAs. As this region showed the strongest signal in a small-scale analysis (*Figure 3—figure supplement 1A*), this region has also been shown to have an unusual codon distribution in previous work (*Tuller and Zur, 2015*; *Goodman et al., 2013*) as explained in the main text. We obtained ncRNA annotations using the Rfam database (version 11.0) (*Gardner et al., 2011*) for the well conserved and highly expressed tRNA, rRNA, RNase P RNA, SRP RNA, tmRNA and 6S RNA families (Rfam accessions: RF00001, RF00005, RF00010, RF00011, RF00013, RF00023, RF00169, RF01854, RF00177). The redundant annotations were filtered for overlapping and identical paralogous sequences, leaving 99,281 core ncRNA that spanned 1582 bacterial genomes.

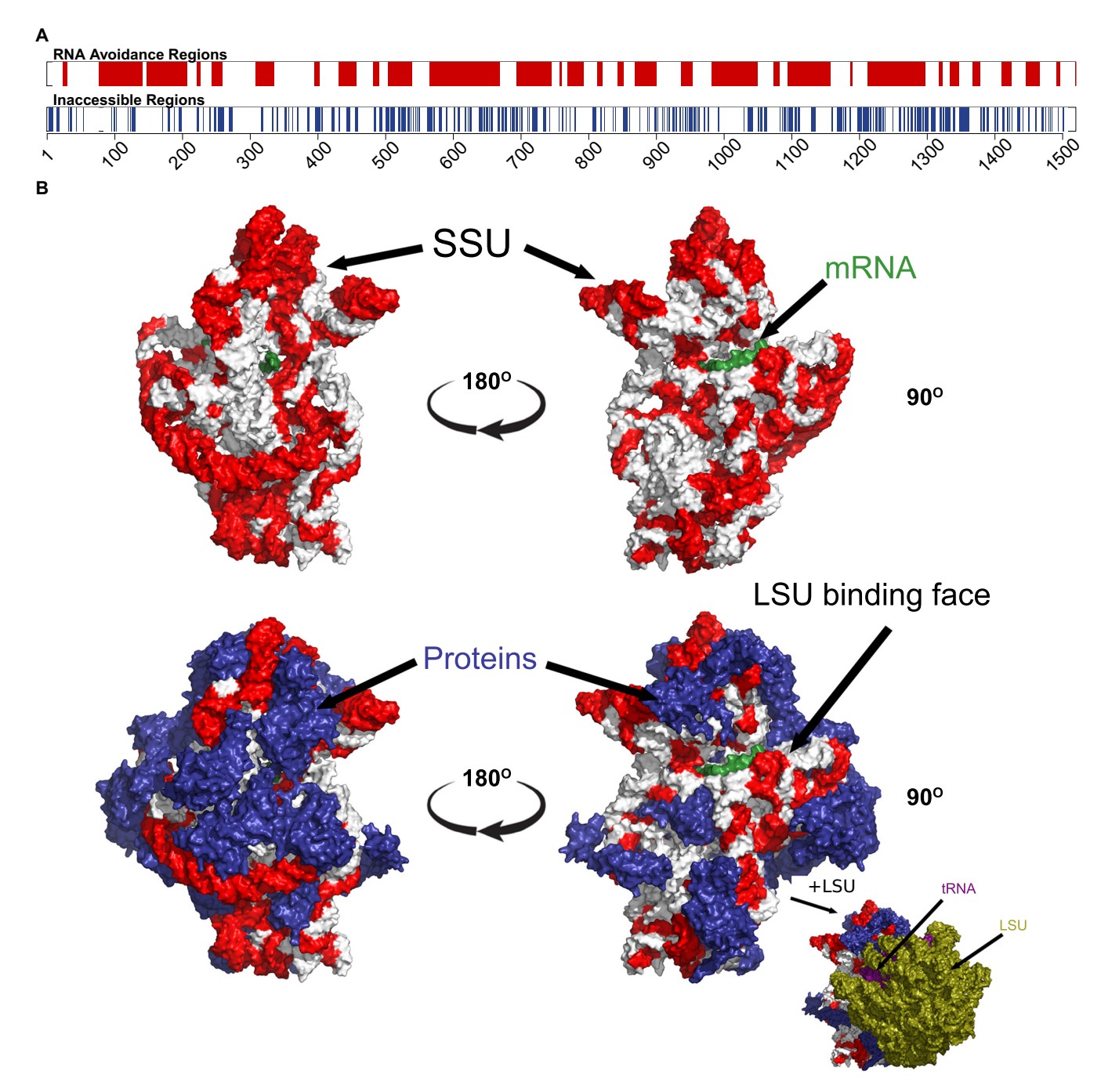

**Figure 3.** The most under-represented mRNA:rRNA interactions correspond to exterior regions of the ribosome. (**A**) In the upper bar, the regions of the *T. thermophilus* SSU rRNA that are under-represented in stable interactions with mRNAs (p<0.05) are highlighted in red. In the lower bar, the inaccessible residues (<3.4 Angstroms from other nucleotides or amino acids in the PDB structure 4WZO). (**B**) The 3 dimensional structure of the *T. thermophilus* ribosome includes 5S, SSU and LSU rRNA, 48 ribosomal proteins, 4 tRNA and a bound mRNA (PDB ID: 4WZO) (*Rozov et al., 2015*). We have highlighted the most avoided regions of the SSU rRNA in red (based upon the fewest stable interactions with *T. thermophilus* mRNAs (p<0.05). Two different orientations are shown on the left and right, the upper structure shows just the SSU rRNA and mRNA structures, the lower includes the ribosomal proteins (coloured blue). Bottom right, a view of the ribosome that also includes the LSU rRNA (green) is also shown. There is a significant correspondence between the accessibility of a region of SSU rRNA and the degree to which it is avoided (p=2.5 × 10$^{-17}$, Fisher's exact test).

The following figure supplements are available for figure 3:

**Figure supplement 1.** Avoidance pattern and its correlation with protein expression vary on mRNAs.

*Figure 3 continued on next page*

*Figure 3 continued*

**Figure supplement 2.** Comparison of different regions for evolutionary conservation analyses.

**Figure supplement 3.** The most avoided regions of selected *T. thermophilus* non-coding RNAs.

## Data sources for archaeal genomes

We followed a similar pipeline for archaeal genomes as described for bacterial genomes. In total we processed 240 archaeal genomes, and after filtering those that had fewer than 90% of the marker genes, we had 118 archaeal genomes for further analysis (genomes available on August 2013) (http://www.ebi.ac.uk/genomes/archaea.html). These genomes host 12,370 and 10,804 core mRNAs and core ncRNAs respectively.

## Test of an (extrinsic) avoidance model

We used RNAup (version 2.0.7) (*Lorenz et al., 2011*) to calculate the binding minimum (Gibbs) free energy (MFE) values of mRNA:ncRNA interactions. The RNAup algorithm combines the intramolecular energy necessary to open binding sites with intermolecular energy gained from hybridization (*Mückstein et al., 2006*). In other words, this approach minimizes the sum of opening intramolecular energies and the intermolecular energy (*Figure 1—figure supplement 1C*). In our model of avoidance, we test for a reduction in absolute binding MFE relative to negative controls as a measure of avoidance. After testing a variety of negative controls (e.g. dinucleotide preserved shuffled mRNAs, the 5′ end of homologous mRNAs from a different bacterial phylum, 100 nucleotides downstream of designated interaction region, reverse complements, and identically sized intergenic regions), we selected the dinucleotide frequency preserved shuffled sequences as our negative control since this displayed the most conservative interaction MFE distribution (*Figure 3—figure supplement 1A–C*). In more detail, to serve as a negative control we compute the interaction MFE between each of the core ncRNAs and 200 dinucleotide-preserved shuffled versions of the 5′ end mRNAs. A dinucleotide frequency preserving shuffling procedure is used, as Gibbs free energies are computed over base pair stacks, i.e. a dinucleotide alphabet, therefore this method has been shown to be important in order to minimise incorrect conclusions (*Workman, 1999*). We tested if the energy difference between native and shuffled interaction distributions is statistically significant using the nonparametric one-tailed Mann-Whitney U test, which returns a single p-value per genome (*Figure 1C*). If the distribution of native interaction energies for a genome is significantly higher (i.e. fewer stable interactions) than the negative control, this is an indication that the genome has undergone selection for mRNA:ncRNA avoidance. To create the background density difference lines (seen in grey at *Figure 1B*), we randomly selected 100 bacterial strains and plot differences between the densities of shuffled interactions.

## Test of an intrinsic avoidance model

The energy-based avoidance model that we defined above is opaque to cases of 'intrinsic avoidance'. These are where the intrinsic properties of mRNA and ncRNA sequences restrict their ability to interact. For an extreme example, if ncRNAs are composed entirely of guanine and cytosine nucleotides, whilst mRNAs are composed entirely of adenine and uracil nucleotides, then these will rarely interact. Therefore, our energy-based avoidance measures for native and shuffled interactions will both be near zero, and thus will not detect a significant energy shift between the native and control sequences. In order to account for some of these issues, we compared the G+C difference between core ncRNAs and core mRNAs. We used a nonparametric two-tailed Mann-Whitney U test to determine if there is a statistically significant G+C difference between the two samples: G+C of ncRNAs vs G+C of 5′ end mRNAs (*Figure 1D,E*).

## Sliding window analysis to detect regions of significance for avoidance on SSU ribosomal RNA

We hypothesise that heterogeneous signals of avoidance within ncRNA sequences may correspond to the accessibility of different ncRNA regions. For example, are highly avoided regions of abundant ncRNAs more accessible than those that are avoided less? To create an avoidance profile, we tested binding MFEs of native and shuffled interactions throughout the full-length SSU ribosomal RNA of *T. thermophilus*, using a one tailed Mann-Whitney U tests to evaluate the degree of avoidance for each nucleotide in the SSU rRNA (*Figure 3*) with a window size of 10 and step size of 1 (*Supplementary file 4*). We selected the protein data bank (PDB) entry (4WZO) as it is one of the few ribosomal structures with associated protein, mRNA, tRNA and LSU binding data (*Rozov et al., 2015*). The native interactions are the interactions between *T. thermophilus* core mRNAs and SSU ribosomal RNA. The shuffled controls are derived from 200 dinucleotide preserved shuffled versions of the RNAs. We created a 2 × 2 contingency table which separates the counts of residues that either host a strong avoidance signal or little avoidance signal (regions with p<0.001, Mann-Whitney U test) and residues that we predict to either be in contact (<3.4 Angstroms between atoms) with ribosomal proteins or ribosomal, transfer or messenger RNAs or not in contact with other molecules (i.e. accessible) (*Figure 3*). We applied a Fisher's exact test *Fisher (1992)* to these groups to and discovered a statistically significant relationship between avoidance and accessibility (p=2.5 × 10$^{-17}$).

We have applied the same analysis to the other *T. thermophilus* core ncRNA genes (tRNAs, tmRNA, RNase P RNA and SRP RNA) in order to determine regions of avoidance (*Figure 3—figure supplement 3*). Since there are more than one tRNAs, we aligned the cellular RNAs to the associated Rfam model (RF00005) (*Gardner et al., 2011*; *Nawrocki et al., 2015*) using the cmalign tool (*Nawrocki and Eddy, 2013*).

## Sliding window analysis to detect regions of significance for avoidance on mRNAs

In order to identify a region of mRNA that is consistent and unique in the datasets that we applied evolutionary and expression analyses to we created an avoidance profile from the previously published GFP mRNAs (*Kudla et al., 2009*). We calculated binding MFEs using a window size of 21 with a 1 nucleotide step size, and for each region we computed the associated Spearman's correlation coefficients with p-values. This analysis revealed the significance of the first 21 nucleotides on expression, this is consistent with previous results that identify initiation as the rate limiting step for translation (*Tuller and Zur, 2015*; *Plotkin and Kudla, 2011*). It also revealed other statistically significant regions with high correlation correlation coefficient throughout the GFP mRNAs (*Figure 3—figure supplement 1A*).

## Proteomics/Transcriptomics and GFP expression

We predict that mRNAs with low avoidance values will produce fewer proteins for each mRNA transcript than those with high avoidance. In order to test this, we conducted a meta-analysis of proteomics and transcriptomics data and the relationship between this data and measures of mRNA and ncRNA avoidance. In the following section we describe the origins of the data we have used and the statistical analysis we use to test whether avoidance influences gene expression.

## Data sources and statistics for mRNA, protein abundance and GFP expression

We compiled our data from five protein and mRNA quantification datasets, which consist of three *E. coli* (*Laurent et al., 2010*; *Goodman et al., 2013*; *Lu et al., 2007*) and two *P. aeruginosa* (*Laurent et al., 2010*; *Kwon et al., 2014*) (*Supplementary file 3*). We calculated Spearman's correlation coefficients (and associated p-values) among the protein abundances and 5´ end secondary structure (measured by intermolecular MFE), codon bias (measured by codon adaptation index (CAI)) and avoidance (*Figure 2A*). We have created single and multiple regression models to determine the explained variances by these parameters (*Figure 2—figure supplement 3* and *Supplementary file 5*). These models show that avoidance explains more variance on average than secondary structure or codon bias. Up to 70 percent of the variation in GFP expression can be explained by including all the parameters and mRNA abundances (*Figure 2—figure supplement 3*).

CAI metric defines how well mRNAs are optimised for codon bias (*Sharp and Li, 1987*). The CAI values were determined based on codon distribution patterns acquired from the core protein coding genes of *E. coli* BL21(DE3) (Accession: AM946981.2) (*Wu et al., 2013*) using Biopython libraries (version 1.6) (*Cock et al., 2009*). The folding MFE predicts how stable the secondary structure of an RNA can be. The folding MFEs of GFP mRNAs were calculated using the RNAfold algorithm (version 2.0.7) (*Lorenz et al., 2011*). We restricted folding energy to first 37 nucleotides because the most significant correlation was previously reported for this region (*Kudla et al., 2009*). We acquired previously published GFP data, associated fluorescence values and mRNA quantifications (*Kudla et al., 2009*) via personal communication. Our avoidance model showed the highest and most significant correlation with GFP expression in that dataset ($R_s = 0.65$, $p=1.69 \times 10^{-20}$) (*Figure 2A* and *Figure 2—figure supplement 2D,E,F*). 5′ end secondary structure ($R_s = 0.62$, $p=5.73 \times 10^{-18}$) correlates slightly less than avoidance, while CAI does not correlate significantly ($R_s = 0.02$, $p=0.4$).

## mRNA design

We have shown that avoidance is a broadly evolutionary conserved phenomenon and that it is significantly correlated with protein abundance relative to mRNA abundance. We now wish to test if avoidance can be used to design mRNA sequences that modulate the abundance of corresponding protein in a predictable fashion. We use a set of GFP mRNA constructs that all maintain the same G+C content, codon adaptation index (CAI) and internal secondary structure but host either very high or very low avoidance values. This procedure was repeated for the CAI and internal secondary structure values while maintaining a constant avoidance. The resulting 13 constructs were synthesised, transformed and expressed by commercial services. In the following paragraphs, we explained how we design our GFP constructs, the experimental set-up and statistical analyses.

## Green fluorescence protein (GFP) mRNA design

We sampled 537,000 synonymous mRNA variants of a GFP mRNA (the 239 AA, 720 nucleotide long, with accession AHK23750, can be encoded by $7.62 \times 10^{111}$ possible unique mRNA variants). In brief, these mRNA variants were scored based upon (1) CAI, (2) mRNA secondary structure in their 5′ end region, and (3) mRNA:ncRNA interaction avoidance in their 5 end region. The genome of *E. coli* BL21 encodes 52 unique core ncRNAs (*Gardner et al., 2011*; *Nawrocki et al., 2015*), to estimate the level of ncRNA avoidance for each GFP mRNA, we sum the binding MFEs. For example, for each GFP mRNA we compute 52 independent binding MFE values for each ncRNA. In short, a higher summed MFE score for a GFP mRNA implies a higher avoidance, while a lower summed MFE score implies a lower avoidance. This approach assumes that the ncRNAs are expressed at much higher levels than GFP mRNAs (i.e. [ncRNA] >> [mRNA]) (*Figure 4*). Consequently, any potential interaction site on GFP mRNAs are likely to be saturated with ncRNA. Finally, we selected 13 GFP mRNA constructs, while controlling the range of G+C values. These GFP mRNAs were designed to have four different aspects; extreme 5 end secondary structure (2 minimum and 2 maximum folding MFE constructs), extreme codon bias (2 maximum and 2 minimum CAI constructs), extreme interaction avoidance (2 minimum and 2 maximum binding MFE constructs) and an 'optimal' construct. The optimal construct was selected for a high CAI, low 5′ end structure and high avoidance. All extreme GFP mRNA constructs have near identical G+C content (between 0.468–0.480) and identical G+C contents at the 5′ end (0.48). Each of the sampled GFP mRNAs is separated from other mRNAs by at least 112 nucleotide substitutions and 122 nucleotide substitutions on average (*Figure 2—figure supplement 1*).

## Extreme GFP transformations, determining fluorescence levels and RT-qPCR analyses

Both GFP expression assays and RT-qPCR analyses were performed as part of a commercial service offered by the University of Queensland, Protein Expression Facility and Real-Time PCR Facility. Plasmid DNA from each construct was transformed into an expression strain of *E. coli* BL21(DE3). Starter cultures were grown in quadruplicate from single colonies in 0.5 mL of TB kanamycin 30 μg/mL media in a 96 deep-well microplate and incubated at 30°C, 400 rpm (3 mm shaking throw). Each starter culture was used to inoculate 1.0 mL of the same media at a ratio of 1:50, each in a single well of a 96 deep-well plate. The cultures were incubated at 30°C, 400 rpm for 1 hr, at this point the

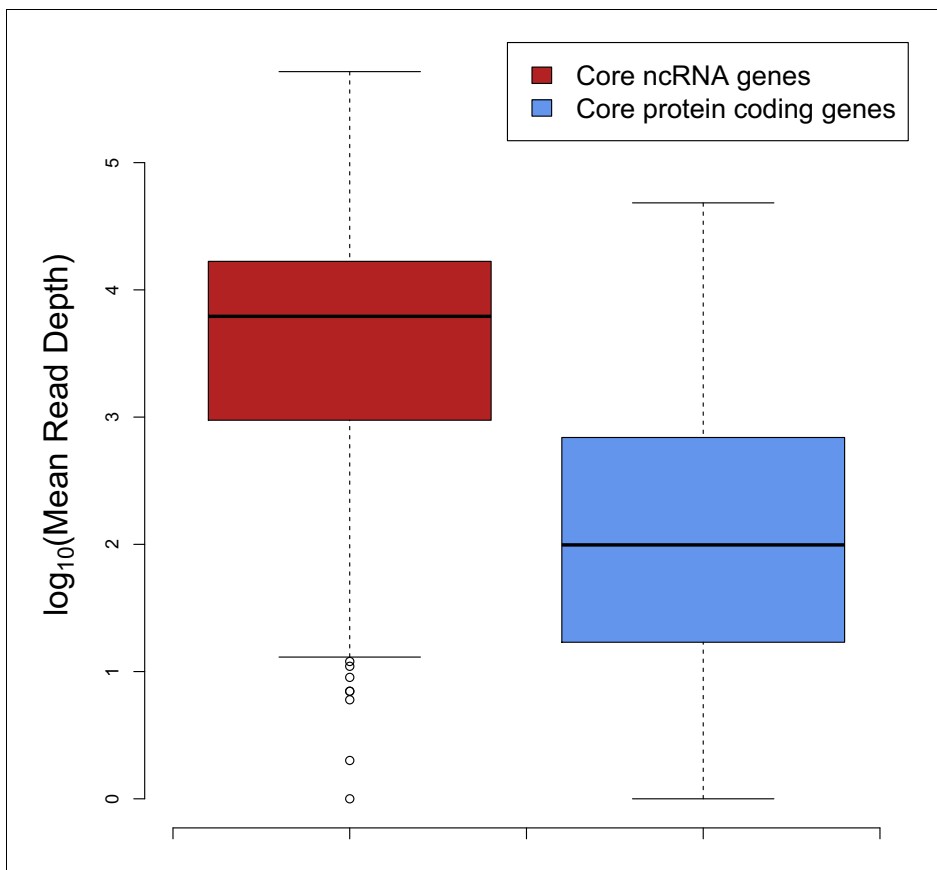

**Figure 4.** The median expression of core ncRNA genes (n = 325 data points) in prokaryotic genomes is nearly two orders of magnitude greater than core mRNAs (n = 8086 data points) which proves that ncRNAs constitute most of the cellular RNAs. To create this plot, we used mean mapped reads per gene length (i.e. mean read depth per position) of each core gene. The expression data are compiled from 5 archaeal and 37 bacterial strains from a previous study (*Lindgreen et al., 2014*).

cultures were chilled for 5 min then induced into 0.2 mM IPTG and incubated at 20°C. For analysis, culture samples of 100 μL were taken at 1 hr, 2 hr, 3 hr, 4 hr and 22 hr (overnight) hours post-induction (HPI) for fluorescence and optical density analysis. Samples were collected in PetriWell 96-well flat bottom, black upper, lidded microplates (Genetix). Cell density of fluorescence measurements was performed on a Spectramax M5 Microplate Reader using SMP software v 5.2 (Molecular Devices). For fluorescence intensity measurements, samples were collected in the 96-well plate listed above. Samples were analysed by bottom-read, 10 reads per well at an excitation wavelength = 488 nm, emission wavelength = 509 nm with an automatic cut-off at 495 nm and measured as relative fluorescence units (RFU). The raw RFU values were normalised by subtracting the averaged baseline values obtained from untransformed BL21(DE3) at the same time point. All samples at the 22 HPI time point were diluted 1:4 in TB kanamycin 30 μg/mL media before measurement. Total RNA was purified from induced 0.5 mL of BL21(DE3) cultures on Maxwell 16 robot (Promega) using LEV simplyRNA Tissue Kit (Promega). RNA concentrations were assessed on Qubit 3.0 Fluorometer (Thermo Fisher Scientific). cDNA synthesis was done using ProtoScript II First Strand cDNA Synthesis Kit (NEB) according to manufacturer protocol using random primer. The rpsL gene was selected as the reference gene (internal control). RT-qPCR was performed in 384-well plates with a ViiA 7 Real-Time PCR System (Thermo Fisher Scientific) using Life Technology SYBR Green-based PCR assay. The data analysis was performed using Applied Biosystems QuantStudio software (Thermo Fisher Scientific). The total volume of reaction was 10 μL including 0.2 μM of each primer as a final concentration. The following PCR conditions were used: 95°C for 10 min, followed by 40 cycles of 95°C for

15 s and 60°C for 1 min. The melting curves were analyzed at 60–95°C after 40 cycles. RNA concentrations were subsequently estimated using the approach (*Schmittgen and Livak, 2008*). We shared the raw data, oligos and primers in the supplementary files (*Supplementary file 2A,B*).

## Statistical analyses of extreme GFP data

As described, we designed extreme GFP mRNA constructs, and measured the associated fluorescence. A Kruskal-Wallis test (nonparametric alternative of ANOVA) shows a statistically significant difference between the fluorescence of GFP mRNA groups (p=1.35 × 10$^{-5}$) (*Figure 2—figure supplement 1*). Our pairwise comparison of GFP groups using a Kruskal-Nemenyi test (a nonparametric alternative of the Student's t-test) for fluorescence difference also reveals a statistically significant difference in fluorescence between high avoidance constructs and low avoidance constructs (p=0.00036). We computed the Spearman's correlation coefficients (and associated p-values) between GFP expression and each of the following measures; CAI ($R_s$ = 0.29, p=0.016), intramolecular folding energy ($R_s$ = 0.34, p=0.006), avoidance (intermolecular binding energy) ($R_s$ = 0.29, p=6.9 × 10$^{-6}$) and mRNA concentration ($R_s$ = 0.73, p=3.2 × 10$^{-3}$) to predict effect size of each predictor. Our avoidance model resulted in the highest correlation with GFP expression (*Figure 2B–D*).

## Acknowledgements

Thanks to Grzegorz Kudla for sharing the GFP expression data from *Kudla et al. (2009)* and Cindy Chang, Emilyn Tan, Michael Nefedov from the RT-PCR and the PEF facilities at the University of Queensland for assistance with generating GFP expression data. We also acknowledge Jeppe Vinther, Lukasz Kielpinski, Anders Krogh and the attendees of the 2012 and 2015 Benasque RNA conference for stimulating discussions.

## Additional information

### Funding

| Funder | Grant reference number | Author |
| --- | --- | --- |
| Royal Society of New Zealand | | Sinan Uğur Umu<br>Anthony M Poole<br>Renwick CJ Dobson<br>Paul P Gardner |
| University of Canterbury | Biomolecular Interaction Centre, Joint PhD Scholarship | Sinan Uğur Umu |
| Royal Society of New Zealand | RutherfordDiscovery Fellowships | Anthony M Poole<br>Paul P Gardner |
| Army Research Office | | Renwick CJ Dobson |

The funders had no role in study design, data collection and interpretation, or the decision to submit the work for publication.

### Author contributions

SUU, PPG, Conception and design, Acquisition of data, Analysis and interpretation of data, Drafting or revising the article, Contributed unpublished essential data or reagents; AMP, Conception and design, Analysis and interpretation of data, Drafting or revising the article; RCJD, Conception and design, Drafting or revising the article

### Author ORCIDs

Sinan Uğur Umu, http://orcid.org/0000-0001-8081-7819
Paul P Gardner, http://orcid.org/0000-0002-7808-1213

## Additional files

**Supplementary files**

• Supplementary file 1. The levels of extrinsic and intrinsic avoidance for each publicly available bacterial and archaeal genome sequence. Column A contains the species and strain names; Column B contains the ENA accession; Column C contains the phylum name; Column D contains the extrinsic avoidance p-value (i.e. the difference between RNAup Gibbs' free-energy of interaction distributions between native and randomized sequences, one-tailed Mann-Whitney U test); Column E contains the genomic G+C content; Column F contains the genome size in nucleotides; Column G contains the average G+C content of the first 21 nucleotides of 114 'core' (highly conserved) mRNAs; Column F contains the average G+C content of six core highly expressed ncRNAs (tRNA, rRNA, RNase P RNA, SRP RNA, tmRNA and 6S RNA); Column F contains the intrinsic avoidance p-value (i.e. the difference in G+C content between core mRNAs and ncRNAs, two-tailed Mann- Whitney U test). Table A contains the data for bacterial species, Table B contains the data for archaeal species.

• Supplementary file 2. Data for the extreme GFP transformations, including fluorescence levels and RT-qPCR analyses. Table A: Column A contains a unique identifier for each mRNA sequence; Column B contains a unique numeric code for each mRNA sequence; Column C contains a colour used for visualizing the datasets; Column D contains the sum of RNAup Gibbs' free-energy of interactions between nucleotides 1 to 21 of the GFP mRNA and the core ncRNAs of *E. coli* BL21(DE3); Column E contains the RNAfold Gibbs' free-energy of folding for nucleotides 1 to 37; Column F contains the CAI for each GFP mRNA, using codon distribution patterns acquired from the core protein coding genes of *E. coli* BL21(DE3); Column G contains the GFP flourescence values (four replicates for each mRNA); Column H contains $\Delta\Delta C_T$ values from RT-qPCR results, these can be used to quantify mRNA concentrations; Column I contains mean GFP fluorescence values; Column J contains mRNA abundances; Table B: contains the oligonucleotide sequences used to gather RT-qPCR results.

• Supplementary file 3. Data corresponding to the heatmap in *Figure 2A*. Each row corresponds to a different dataset containing protein and mRNA expression levels. Column A contains a brief summary of the type of dataset; Column B contains Spearman correlation coefficients (and corresponding p-values) between avoidance measurements and protein abundance; Column C contains Spearman correlation coefficients (and corresponding p-values) between mRNA secondary structure measurements measurements and protein abundance; Column D contains Spearman correlation coefficients (and corresponding p-values) between codon adaptation index measurements and protein abundance; Column E contains Spearman correlation coefficients (and corresponding p-values) between mRNA and protein abundances; Column F contains the species names of the organism each dataset was collected in; Column G contains the size of the dataset; Column H contains a reference to the manuscript each dataset was first published in; Column H contains a link to the Pubmed entry for each manuscript;

• Supplementary file 4. Data corresponding to *Figure 3*. <3.4 For each position in the SSU rRNA sequence provided in PDB structure 4WZO, in Column A the accessibility of the region is evaluated (i.e. '1' if the residue is Angstroms from another nucleotide or aminoacid residue and '0' otherwise). Column B contains local SSU:mRNA avoidance p-values corresponding to differences in RNAup Gibbs' free-energy of interaction distributions between native and randomized sequences using Mann-Whitney U tests.

• Supplementary file 5. This table contains the results of $R^2$ values for linear regression models between protein levels and the different predictors of expression level that are illustrated in *Figure 2—figure supplement 3*. Column A contains the model (e.g. "Protein abundance Avoidance + Folding Energy + CAI + mRNA abundance" corresponds to the relationship between protein abundance, mRNA avoidance, mRNA folding, codon usage and mRNA abundance; Column B contains the $R^2$ value for the relationship; Column C contains the reference the corresponding protein and mRNA data came from; Column D contains the type of data and the size of the dataset;

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
