## [Decision Letter]

Thank you for submitting your article "Avoidance of stochastic RNA interactions can be harnessed to control protein expression levels in bacteria and archaea" for consideration by *eLife*. Your article has been reviewed by three peer reviewers, one of whom is a member of our Board of Reviewing Editors, and the evaluation has been overseen by Naama Barkai as the Senior Editor. The reviewers have opted to remain anonymous.

The reviewers have discussed the reviews with one another and the Reviewing Editor has drafted this decision to help you prepare a revised submission.

Summary:

This manuscript from Umu et al. provides evidence that stochastic base pairing between highly abundant non-coding RNAs (ncRNAs) such as tRNAs, rRNAs etc. and sequences within the 5'end of messenger RNAs (mRNAs) can be deleterious to the translation of the mRNAs. They suggest that this base pairing causes some of the recognized disconnect between mRNA and protein levels, and that these stochastic interactions can have a bigger effect on protein production than either mRNA structure or codon bias. They propose a factor called 'avoidance', which refers to the potential of sequences within the first 21 nt of the mRNA to hybridize to ncRNAs, with lower potential for interaction yielding higher overall expression of the gene product.

This is an interesting premise, and if fully fleshed out, could provide new insights into mRNA sequence evolution (especially of the 5'-UTR) and an additional explanation for the imperfect correlation between mRNA and protein levels. Overall, the data indicating that interactions between ncRNAs and mRNAs are avoided and could repress translation are generally convincing. Careful controls are employed for the computational and experimental studies.

However, the reviewers found that the data were largely correlative at this stage, and lacking in development of specific examples of avoidance (or the lack thereof) affecting mRNA expression. The point was also raised that small non-coding RNA molecules that are not tightly bound by proteins would be better case studies than the SSU rRNA.

Thus, we invite a revision that addresses these issues specifically, as described in more detail below.

Essential revisions:

1) The authors stopped short of providing the most interesting and important result: showing that they could better explain the relationship between mRNA levels and protein levels upon consideration of avoidance. The authors use this as motivation for their studies, but don't bring their story full circle and show that integrating avoidance into the equation helps us understand the non-uniform relationship between mRNA and protein. They find that avoidance is correlated with higher protein expression in several data sets, but fail to dig deeper. Without this demonstration of utility, we are concerned about how broad the readership for this manuscript will be.

a) The authors should investigate deeper into where native mRNAs with particularly high or low avoidance scores fall on the spectrum of mRNA level vs. protein level.

b) They should attempt to define how much better proteome predictions could be if taking avoidance into account.

2) The rationale for including some of the well-conserved ncRNAs such as RNase P RNA and 6S RNA that are generally tightly bound by proteins and largely unavailable for RNA-RNA interactions is not clear. An example of an abundant non-coding species that was freely available for stochastic interactions would be stronger.

3) The authors should present the specifics for several mRNAs to illustrate the impact of the three parameters.

---

## [Author Response]

*Essential revisions:*

*1) The authors stopped short of providing the most interesting and important result: showing that they could better explain the relationship between mRNA levels and protein levels upon consideration of avoidance. The authors use this as motivation for their studies, but don't bring their story full circle and show that integrating avoidance into the equation helps us understand the non-uniform relationship between mRNA and protein. They find that avoidance is correlated with higher protein expression in several data sets, but fail to dig deeper. Without this demonstration of utility, we are concerned about how broad the readership for this manuscript will be.*

We have added results from using linear models combining avoidance, internal secondary structure, codon usage and messenger RNAs. The R^2^ when combining all the parameters range from 0.7 (our data) to 0.2 (Goodman et al. data). By removing the avoidance measure from the linear model the R^2^ values decrease between 19% and 56% from the R^2^ values in the GFP datasets. In the mass-spec data the R2 values decrease between 3 and 0.2%. Compare this with when we remove internal secondary structure information the R^2^ values decrease between 18% and -1% for GFP data, and between 1.5% and -0.02% for the mass-spec data. When we remove codon usage statistics the R^2^ values decrease between -0.05% and -0.4% for GFP data, and between 9% and -0.06% for the mass-spec data. This is consistent with the avoidance model being at least as good an explanation for variation in protein expression as either internal secondary structure and codon usage statistics. The manuscript has been updated to reflect some of this new analysis. See Figure 2—figure supplement 3 in the new manuscript.

*a) The authors should investigate deeper into where native mRNAs with particularly high or low avoidance scores fall on the spectrum of mRNA level vs. protein level.*

For one of the datasets we have investigated the distribution of protein-per-mRNA ratios (log[protein-per-mRNA]) (Laurent et al. 2010). We have taken the top and bottom 10 most abundant or scarce genes based upon this metric. Then for each of the codon usage, internal secondary structure and avoidance measures we have computed Z-scores for the 20 extreme genes (using means and standard deviations derived from the total dataset). We have found that avoidance is extremely high in the group of 10 abundant genes and is extremely low in the 10 scarce datasets. The shift in Z-scores relative to the null distribution is more modest in the other two measures. Figure 2—figure supplement 4 has been added to the manuscript and the discussion has been updated.

*b) They should attempt to define how much better proteome predictions could be if taking avoidance into account.*

See the discussion and the additional figure in response to point 1, above.

We found that avoidance alone can explain more than 36 percent of variation in synthetic mRNA expression. Avoidance outperforms the other two global factors in synthetic mRNA expression.

However, it can only explain around 2 percent of variation in native mRNA expression, which still makes avoidance a comparable factor with codon usage and secondary structure.

It is plausible to accept that native mRNAs are already selected for their high avoidance, which we detected as a conserved avoidance signal in core mRNA genes. A similar observation is also true for secondary structure. It also marginally influences protein expression in synthetic mRNA expression, but does not have similar effect on native mRNA expression.

*2) The rationale for including some of the well-conserved ncRNAs such as RNase P RNA and 6S RNA that are generally tightly bound by proteins and largely unavailable for RNA-RNA interactions is not clear. An example of an abundant non-coding species that was freely available for stochastic interactions would be stronger.*

There are relatively few ncRNA genes that we can say are never bound by protein and are broadly conserved across the range of species that we have analysed. There is increasing evidence that evolutionary turnover of ncRNA genes is rapid (Lindgreen et al. 2014). Furthermore, during assembly and degradation of ribonucleoparticles, presumably most components of these abundant ncRNAs will be accessible at some time. The tRNAs may be the least bound, by proteins (other than a transient interactions with tRNA synthetases and the ribosome). Furthermore, even for highly structured and bound ncRNAs such as the SSU rRNA, we see strong signatures of avoidance. We have included the below text in the manuscript to hopefully make this justification clearer:

In order to ensure that our analysis is comparable across all bacteria and archaea we have focussed on just the most highly conserved ncRNA and protein-coding genes. Although, many of the ncRNAs are highly structures and are bound by RNA-binding proteins this is not the case during either synthesis and degradation of these products, furthermore, a fraction of the RNA components of these genes will be exposed. Therefore we expect these will form useful datasets for initial testing of our hypothesis.

We have also included an additional figure (Figure 3—figure supplement 3) that shows the regions of *T. thermophilus* tRNAs, tmRNA, RNase P and SRP RNA that have the fewest interactions with mRNAs (i.e. are the most avoided).

*3) The authors should present the specifics for several mRNAs to illustrate the impact of the three parameters.*

The outlier analysis described in response to 1a) hopefully addresses this point.